# Evaluation of Temporary Urethral Stents in the Management of Malignant and Nonmalignant Urethral Diseases in Dogs

**DOI:** 10.3390/vetsci9020063

**Published:** 2022-02-01

**Authors:** Jody P. Lulich

**Affiliations:** Department of Veterinary Clinical Sciences, University of Minnesota, St. Paul, MN 55108, USA; lulic001@umn.edu; Tel.: +1-612-625-7277

**Keywords:** canine, transitional cell carcinoma, urethral obstruction, urinary catheter, urinary incontinence, urinary tract

## Abstract

Urethral stent placement is a minimally invasive interventional procedure commonly performed by specialists to alleviate urethral obstruction. However, the availability of urethral stents is limited by their high cost and the need for special equipment. The aim of this retrospective study was to describe the construction and placement of an inexpensive temporary urethral stent and to report on its outcome in managing dogs with naturally occurring urethral disease. Temporary stents were placed in the urethra of 17 dogs with malignant and nonmalignant urethral pathologies. The most common indication for temporary stent placement was urethral obstruction. In this population, urethral patency was restored in all dogs. The most frequently reported complication was urinary incontinence. To manage this complication, dogs were diapered. Temporary urethral stents served as a durable and inexpensive alternative to expanding metallic stents. Because temporary stents were constructed from readily available materials and inserted without special equipment, practitioners capable of catheterizing the urethra can insert them on demand.

## 1. Introduction

Among the various indications for urinary stent placement, the need for immediate relief of urethral obstruction is perhaps the most pressing. While indwelling catheters can offer immediate drainage of the urinary bladder, continuous catheterization represents an externalized foreign body that exposes the urinary tract to increased risk of fecal contamination and nosocomial infection [1,2,3]. When catheters are connected to a closed collection system, the patient becomes tethered to hospitalization, resulting in increased cost and increased exposure to multidrug-resistant microorganisms [4,5]. As an alternative, expandable metallic stents have been successfully placed inside the urethra of dogs with malignant and nonmalignant causes of obstruction [6,7,8,9,10,11]. However, their high expense and the need for specialized training and equipment to insert them minimize availability and delay installation [12]. The purpose of this report was to describe the construction and placement of a temporary urethral stent that was easily manufactured from materials typically available in most veterinary clinics, allowing stents to be inserted on demand as an outpatient procedure. A second goal was to report on the indications and short-term outcomes for their use in dogs with naturally occurring urethral diseases.

## 2. Materials and Methods

### 2.1. Case Selection

Medical records from the Veterinary Medical Center, University of Minnesota, of dogs receiving care between 2005 and 2021 were reviewed. Only records coded for placement of a temporary urethral stent were included. To ensure that all dogs that received a temporary stent were included, no exclusion criteria were applied. From each record, the following data were tabulated: signalment; urinary signs; underlying urethral pathology; and stent placement details, including indication, duration, and short-term outcomes. Records were not evaluated for treatment of the primary urethral disease.

### 2.2. Stent Construction and Placement

Stents were constructed from a flexible, 8Fr urinary catheter with two side drainage holes at the tip (Kendall urethral catheter, Covidien LLC, Mansfield, MA 02048, USA). Catheter length was determined by measuring over a lateral abdominal radiograph the path of the urethra from the middle of the vestibule in females or the tip of the penis in males to 1 cm cranial to the urethral lesion. The catheter was cut to length, retaining the end with the drainage holes as the stent. Parallel and close to the cut surface, a 20 gauge hypodermic needle was inserted through the walls of the stent to create a path to place the suture (Figure 1). A nonabsorbable 3.0 suture was passed into the hollow end of the hypodermic needle and out the other side. The needle was pulled out of the stent, leaving the suture in the needle’s track. The suture was tied to create a 1 to 2 mm loop that was used to anchor the stent inside the vestibule. The suture close to the knot was cut free. The longer cut section was threaded through the small loop, and the ends were tied to make a safety loop attached to the stent. The safety loop was used to adjust the stent’s position in the urethra.

Under deep sedation or general anesthesia, the dog was placed in lateral recumbency. Hair around the vestibule in females or prepuce in males was clipped. The area was aseptically scrubbed and the inside of the vestibule or prepuce was flushed with a 0.04% chlorohexidine solution.

An otoscope was used to visualize the external urethral orifice inside the vestibule. The stent was inserted through the otoscope cone and passed into and down the urethra. Once the stent bypassed the lesion or entered the urinary bladder, urine exited the stent, confirming its location and function.

A suture was used to secure the stent in place. In females, a finger was inserted in the vestibule to the external urethral orifice and elevated laterally to indicate the insertion site. A nonabsorbable suture attached to a cutting needle was passed through the skin and into the lumen of the vestibule in the direction of the external vulvar opening. The vulvar lips may need to be everted to locate the tip of the needle. Once visualized, the needle tip was secured with hemostats. The needle and its attached suture were pulled out of the vestibule. To prevent pulling the suture completely through the skin, a hemostat was clamped on the opposite end.

With the stent already in the urethra, pulling the safety loop exteriorized the distal tip of the stent, exposing its anchoring loop. The needle and suture were passed through the anchoring loop. To secure the stent back down the urethra, the needle with suture was passed in the opposite direction, first through the external opening of the vestibule, then through the inner lateral wall, and finally through the skin exiting near its original entry. The ends of the suture were tied near the skin, but with sufficient laxity so as not to indent the skin. Tying the suture pulled the stent back into the vestibule and down the urethra, securing the distal end of the stent to the inside lateral wall of the vestibule (Figure 2).

In male dogs, the external suture was placed through the lateral prepuce approximately 1 to 2 cm proximal to the external preputial opening and secured in a similar fashion as in females. In one large male dog, a stent constructed from a urinary catheter was too short to accommodate a 60 cm urethral length. In this dog, a 10 Fr nasoesophageal feeding tube (Mila International Inc., Florence, KY 41042, USA) was cut to length, inserted down the urethra, and secured with a suture through the wall of the penis using a groove director in a manner similar to surgical correction of a prolapsed urethra (Figure 3) [13]. Dogs typically received 2 to 4 doses of an antibiotic at recommended intervals during and after stent placement.

Stent placement was inspected with digital palpation, vaginoscopy, or radiography. If appropriately placed, the safety loop was cut and pulled free from the stent.

To remove the stent, a finger, hemostat, or urethral biopsy instrument was inserted inside the vestibule, prepuce, or external urethral orifice and fastened on the stent. The external suture was cut and the stent was pulled out. If needed, topical application of local anesthetics was instilled to maximize patient comfort prior to stent removal.

## 3. Results

Temporary urethral stents were placed in 17 dogs, 13 spayed females and 4 neutered males. In 16 dogs, the indication for stent placement was relief of urethral obstruction: nine with urethral cancer, three with extraurethral cancer compressing the urethral lumen, two with nonmalignant mural urethral swelling occluding the urethral lumen, and two with functional closure of the urethra of undetermined cause. In one dog without urethral obstruction, a stent was placed to divert urine through the stent to facilitate spontaneous healing of a urethral tear.

The length of the stent traversed the entire urethra and extended into the caudal urinary bladder in 16 of 17 dogs. In one male dog experiencing urethral obstruction following prescrotal urethrostomy, a short (6 cm) stent was secured within the urethra near the urethrostomy opening. 

Following stent placement, all dogs were able to evacuate urine out of the urinary bladder. The temporary stent was well tolerated; no dog needed an E-collar or other apparatus to deter stent removal. All dogs exhibited continuous urinary incontinence except the one male with the 6 cm stent placed in the distal membranous urethra. This dog displayed normal urination.

Nine dogs with urethral cancer were stented; seven females with transitional cell carcinoma (two Labrador retrievers, two Lhasa apsos, one beagle, one Pembroke corgi, and one mixed breed; age range = 6 to 12 years), one female Labrador retriever with leiomyosarcoma, and one mixed breed male with lymphoma (Table 1). Three dogs were euthanized on days 7, 24, and 105 because of their primary disease; stents remained patent without migration until the time of euthanasia. One dog with a urethral leiomyosarcoma reobstructed on day 57. The stent was no longer in the urethra and the anchoring suture was missing. In two dogs, stents were removed on days 6 and 62 and replaced with a permanent metallic stent; temporary stents were patent and functioning at the time of removal. However, on day 61 in the latter case, the external portion of the anchoring suture had dug underneath the skin but had not completely detached through all tissue layers. This dog received palliative radiation therapy (four treatments of 5 Gy each) with the temporary stent in place. The temporary stent was removed using gentle traction before a permanent expanding metallic stent was placed in the urethra. In a male dog with lymphoma, the stent was placed to relieve obstruction while waiting for biopsy results of the urethral lesion. However, on the same day, the owners elected for a prescrotal urethrostomy and the stent was removed. Two dogs were lost to follow-up. 

Three dogs with extramural (nonurethral) cancer were stented: a 13-year-old female Staffordshire terrier with a caudal abdominal sarcoma, a 10-year-old female beagle with intrapelvic hemangiosarcoma, and a 13-year-old female bichon with anal gland adenocarcinoma. Dogs were euthanized on days 1, 10, and 337, respectively. The urethra remained patent up to the time of euthanasia. However, in the last case, the medical record did not indicate if the stent or the external anchoring suture was in place at the time of euthanasia.

Two dogs were stented because of non-neoplastic urethral inflammation and swelling. An 8-year-old male bichon frise was not able to urinate following traumatic extraction of a urethrolith. The stent was removed on day 5. An 8-year-old male Labrador retriever could not urinate following urethrostomy surgery. Marked swelling at the urethrostomy site occluded the urethra. A short stent (6 cm) was anchored approximately 2 mm within the urethra, proximal to the urethrostomy opening. The stent was removed on day 13. Both stents functioned appropriately without complications. Following stent removal, both dogs urinated normally.

A 9-year-old female boxer was stented to allow spontaneous healing of a urethral tear. The urethra was damaged following removal of a retroperitoneal hemangiosarcoma that surrounded the urethra. The stent was removed on day 9, at which time the dog urinated normally.

Temporary stents were placed in two dogs with functional urethral obstruction. In a 12-year-old female Labrador retriever, a structural cause for obstruction was not identified by contrast urethrocystography. For unknown reasons, the stent detached and migrated out of the urethra on day 13. At that time, the dog reobstructed and was euthanized. Similarly, a 5-year-old St. Bernard with hind limb ataxia and urethral obstruction was stented. The working diagnosis was suprasacral spinal cord disease. The cause for urethral obstruction and hind limb ataxia was not investigated further. The stent was placed as a provocative test to determine if a flaccid, overdistended bladder could regain sufficient contractility to overcome functional urethral closure. The dog also received tamsulosin (0.4 mg q 24 PO orally) to relax the proximal urethra. The stent was in place for 11 days. One day after stent removal, the dog’s urinary signs recurred; he was unable to empty his bladder and was euthanized.

Urine cultures were performed once in four dogs on days 7, 13, 16, and 56 after stent placement. Urine culture results were negative on day 7 in one dog and positive in three other dogs. All samples were obtained via free catch urine; bacterial colony counts were greater than 100,000 per mL of urine except for an *E. coli* positive culture that was 10,000 colonies per mL of urine [14]. The three dogs with positive poststent bacterial cultures were also positive 7 days prior to stent placement (Table 1).

## 4. Discussion

Temporary urethral stents were successfully placed in all 17 dogs. The most common indication for stent placement was alleviation of urethral obstruction. In this population, urethral patency was restored in all 16 dogs. In one dog without urethral obstruction, the stent was placed to divert urine, resulting in natural healing of a urethral tear. 

Temporary urethral stents were constructed from materials typically available in most veterinary clinics and inserted without utilizing medical imaging, cystoscopy, or specialized training, allowing them to be placed on demand and as an outpatient procedure. To facilitate home care, stents were anchored within the urogenital tract to be less accessible to patient removal and unlike indwelling urinary catheters were protected from fecal contamination.

Common concerns with urethral stent placement include urinary incontinence, urinary tract infection, stent migration, and pain [6,7,8,9,10,11]. After placing permanent metallic stents, urinary incontinence occurred in 41% of 17 dogs stented for transitional cell carcinoma, in 45% of 11 dogs stented for nonmalignant causes of urethral obstruction, and in 64% of 42 dogs stented for urethral carcinoma [7,8,9]. In the present study, temporary stents were associated with urinary incontinence in 94% (13 of 13 females and 3 of 4 males). This was expected since temporary stents were placed across the entire urethra, disabling urethral sphincters, in 16 of 17 dogs. To minimize inappropriate urine soiling, incontinent dogs were diapered. In male dogs, only the proximal urethra sphincter is necessary to maintain continence, which may explain why in one case, stenting only the distal urethra preserved continence in that dog [15].

In addition to disabling urinary sphincters, stents and indwelling urinary catheters disrupt host defenses that normally guard against urinary tract infection. As they alter urine flow, serve as a nidus for bacterial adhesion and biofilm production, and stimulate urothelial inflammation, it is critical to know when the urinary tract will become colonized with bacteria and if treatment is necessary [16,17]. In a study of dogs with intervertebral disc disease and indwelling urinary catheters, the odds of a urinary infection increased by 27% for each successive day the urethra was catheterized [18,19]. No studies have prospectively reported the rate of stent-associated urinary tract infection as a function of stent duration in dogs. Only four dogs with temporary stents reported in this study had urine submitted for bacteriological culture after stent placement. Three cultures were positive for bacteria. However, it was not possible to determine if the stent was the primary risk for infection because in each case the urine culture before stent placement was also positive. It is unknown if urinary infection rates would be lower for a temporary stent in a dog residing in its home where bacteria populations are potentially fewer and less virulent compared to maintaining an indwelling urinary catheter in a veterinary intensive care setting [3,20]. Further studies are needed to determine the risk of urinary tract infection in dogs with temporary stents.

Migration of temporary urethral stents occurred in 3 of 17 dogs. In two dogs, stents detached by days 13 and 57 for causes that were not clear. In both dogs, the urethra reobstructed, and both dogs were euthanized. In the third case, the external suture dug beneath the skin but remained anchored inside the vestibule and attached to the stent. It is unknown if the radiation therapy that the dog received to slow the growth of transitional cell carcinoma contributed to the migration of the anchoring suture of the temporary stent. In that dog, the stent functioned properly for up to day 62, at which time it was replaced with a permanent metallic stent. Based on their construction, temporary stents have several areas that could fail and result in migration. These include untying of the suture knot, suture break down, suture migration beneath the skin, and deterioration of the attachment site at the distal end of the stent. Lastly, the stent can flip backward out of the urethra. To recognize potential problems early, stents should be inspected periodically. If migration becomes problematic, stents can be easily removed and replaced.

Stent comfort was not assessed in this study. However, using a narrower, nonexpanding stent would be expected to be more comfortable than a metallic stent that typically expands to a larger diameter [6,7,8,9]. In addition, the temporary stent was loosely anchored to minimize tension on urogenital structures.

In this study, temporary stents were a simple, rapid, and inexpensive method to alleviate urethral obstruction caused by irreversible urethral and periurethral cancer. Inserting a temporary stent provided caregivers additional time to make medical and ethical treatment decisions without the urgent concern of urinary obstruction. Once prognoses were established, some decided to place a more permanent urethral stent while others elected to terminate care and leave the temporary stent in place. 

Indwelling urethral catheters facilitate management of urethral obstruction and recovery from suprasacral intervertebral disk disease, detrusor atony, granulomatous urethritis, detrusor sphincter dyssynergia, and reconstructive urethral surgery [2,18,19,20,21,22,23,24,25,26]. However, indwelling catheters often necessitate hospitalization, particularly when connected to a closed collection system [4,5]. Potential advantages of placing temporary stents for these disorders include improved mobility and reduced hospitalization and fecal contamination of the urinary tract [1,2,3,4]. 

Based on this study, three changes in stent design and placement were considered. A bolster (e.g., button or flexible tubing) can be placed between the external suture and skin to minimize suture migrating below the skin’s surface. If the stent is too flexible or too small to be easily passed down the urethra, it can be fitted over a firmer and narrower lubricated urinary catheter to assist its placement. This change may also facilitate stent insertion for operators who are more experienced at passing catheters without visualization or with digital palpation. If urethral pathology prevents retrograde urethral access, percutaneous antegrade urethral access may be possible [27]. Lastly, because male dogs often exteriorize their penis to urinate, attaching the external anchoring suture to the penis instead of the prepuce seems more appropriate to improve patient comfort.

Limitations of the present study were its retrospective nature and small sample size, which limited the study’s ability to assess infection rate and degree of urethral comfort. Since all stents were placed by a single operator at a single institution, the generalizability of the procedure could be questioned. However, the simplicity of the procedure without the need for specialized equipment or training supports a universal application with similar outcomes irrespective of the operator.

## 5. Conclusions

Temporary urethral stents were a durable, inexpensive, short-term alternative to metallic expanding stents to alleviate urethral obstruction. Because temporary stents are constructed from readily available materials and inserted without special equipment, practitioners capable of catheterizing the urethra can insert them on demand. Unlike expanding metallic stents, temporary stents can divert urine while allowing apposition and re-epithelialization of wound edges from a urethral tear to heal. Urinary incontinence was a common complication of temporary stents. If diapering is not tolerated, the temporary stent can be easily removed and replaced with a permanent metallic stent or other urine diversion treatments associated with less frequent urinary incontinence [28,29,30].

## Figures and Tables

**Figure 1 vetsci-09-00063-f001:**
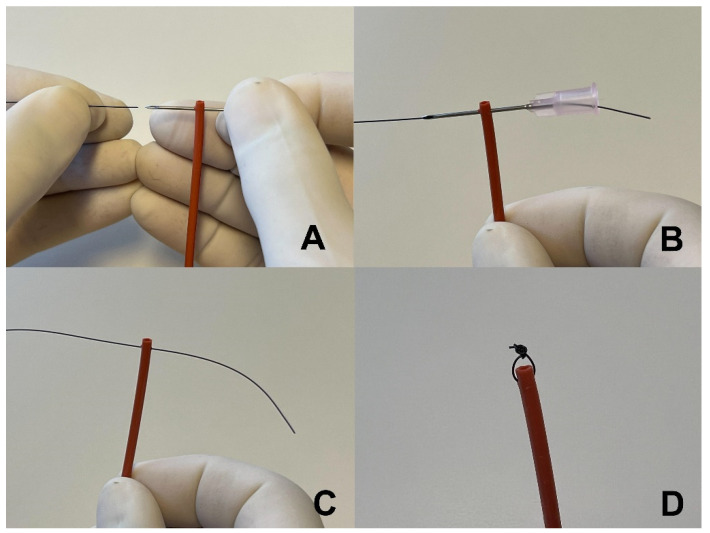
Construction of a temporary stent: (**A**) A hypodermic needle is inserted through the distal end of the stent and a suture is inserted into the end of the needle. (**B**) The suture is passed all the way through the hypodermic needle until it exits the needle’s hub. (**C**) The needle is retracted, leaving the suture in the needle’s track. (**D**) The suture is tied to create a small loop at the distal end of the stent.

**Figure 2 vetsci-09-00063-f002:**
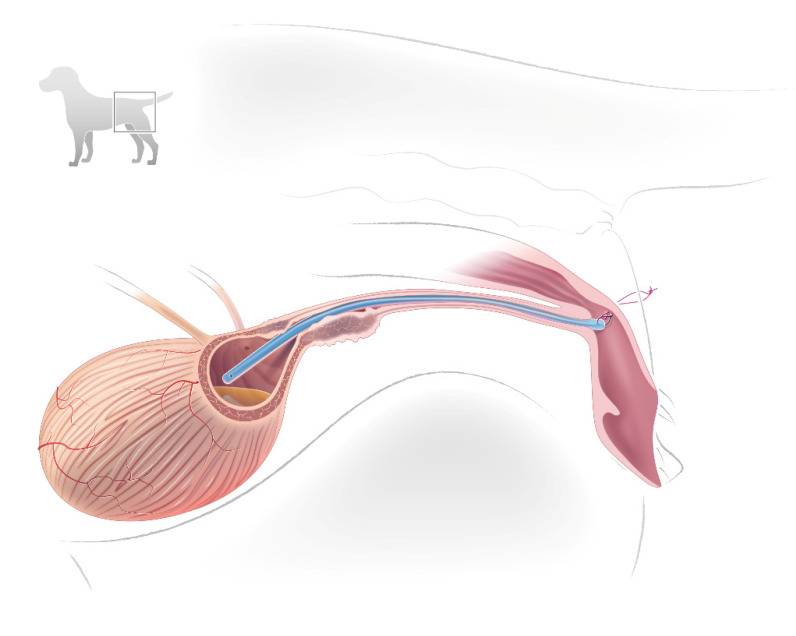
Illustration of a temporary stent placed in the urethra of a female dog. The proximal tip of the stent is placed cranial to the obstructing urethral lesion. The distal tip of the stent is anchored to the inside wall of the vestibule with a loop of suture loosely tied over perineal skin near the vulva. The square in the upper insert identifies the location of interest in the dog. © Diogo Guerra.

**Figure 3 vetsci-09-00063-f003:**
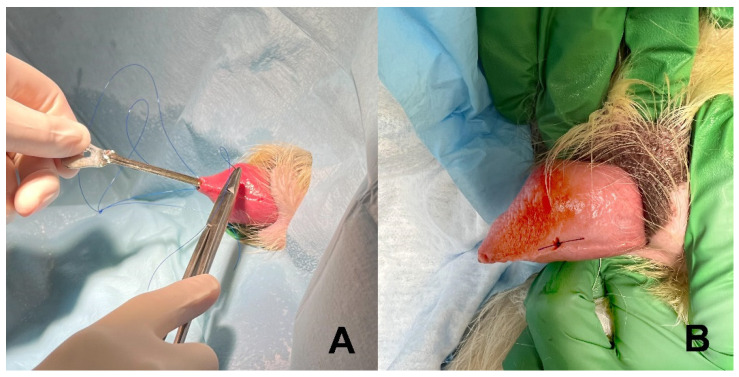
Placing a suture to anchor the temporary stent inside the urethral lumen of a male dog. (**A**) A groove director is inserted in the urethral lumen to assist direction of the needle and suture through the ventral wall of the penis, into the urethral lumen and out the external urethral orifice. (**B**) Completed suture after it was attached to the anchoring loop on the temporary stent, passed back into the urethral lumen and out of the ventral wall of the penis, and tied.

**Table 1 vetsci-09-00063-t001:** Indication, duration, and outcomes of placing temporary urethral stents in 17 dogs with urethral cancer, extraurethral cancer, urethral inflammation, functional urethral obstruction, or a urethral tear.

Dog				Stent					Patient Outcome
Age (years)	Sex	Breed	Urethral Diagnosis	Indication	Duration (days)	Adverse Consequences			
						Incontinence	Migration	Urine Infection	
Urethral Cancer									
11	FS	Labrador	TCC	Obstruction	NFU	Yes	NFU	NFU	NFU
7	FS	Lhasa	TCC	Obstruction	NFU	Yes	NFU	NFU	NFU
1	MN	Golden	Lymphoma	Obstruction	1	Yes	No	NFU *	Urethrostomy
11	FS	Corgi	TCC	Obstruction	6	Yes	No	NFU	Permanent stent
12	FS	Beagle	TCC	Obstruction	7	Yes	No	NFU *	Euthanized
6	FS	Labrador	TCC	Obstruction	24	Yes	No	NFU *	Euthanized
10	FS	Labrador	Leiomyosarcoma	Obstruction	57	Yes	Yes	Enteroc * (day 13)	Euthanized
10	FS	Mixed	TCC	Obstruction	62	Yes	Yes	NFU *	Permanent stent
12	FS	Lhasa	TCC	Obstruction	105	Yes	No	Staph * (day 56)	Euthanized
Extraurethral Cancer									
13	FS	Staffordshire terrier	Abdominal sarcoma	Obstruction	1	Yes	No	NFU	Euthanized
10	FS	Beagle	Intrapelvic Hemangiosarcoma	Obstruction	10	Yes	No	NG (day 7)	Euthanized
13	FS	Bichon	Anal gland adenocarcinoma	Obstruction	337	Yes	?	NFU	Euthanized
Urethral Inflammation									
8	MN	Bichon	Urethral swelling after stone removal	Obstruction	5	Yes	No	NFU	Normal voiding
8	MN	Labrador	Urethral swelling after urethrostomy	Obstruction	13	No	No	NFU	Normal voiding
Functional Urethral Obstruction									
5	MN	St. Bernard	Functional	Obstruction	11	Yes	No	NFU	Urethral obstruction, euthanized
12	FS	Labrador	Functional	Obstruction	13	Yes	Yes	Pseudom * Enterococ * *E. coli* (day 16)	Urethral obstruction, euthanized
Urethral Perforation									
8	FS	Boxer	Urethral tear	Urine diversion	9	Yes	Yes	NFU	Healed

TCC, transitional cell carcinoma; NFU, no follow-up; * positive urine culture within 7 days prior to stent placement.

## Data Availability

All data are reported in this manuscript.

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
