# Peer review of "Evaluation of Temporary Urethral Stents in the Management of Malignant and Nonmalignant Urethral Diseases in Dogs"

_vetsci, 2022, doi:10.3390/vetsci9020063_

Round 1

Reviewer 1 Report

The article deals with a particularly important topic in palliative medicine: it is not always possible to perform surgery on the urethra or bladder at the urethral orifice and these are life treating conditions.

The title is appropriate for the paper and the abstract summarizes well the topic that will be addressed.

As a suggestion I make to the author, the keywords should be words that do not already appear in the title, so that we can broaden the searchability of the article.

The introduction states that the use of the stent for the control of urethral obstruction, compared to indwelling catheterization, is preferable as it reduces the risk of infection and hospitalization but, the available stents are of limited use due to the high cost. Therefore, the author proposes an inexpensive and easy-to-make stent model.

The results are clearly understood. At line 183 it is stated that the analysis was done on “free catch” urine, so this method makes the result non completely reliable.

The materials and methods section clearly describes the technique and principles behind the choice of a temporary stent, and certainly the availability of the material used makes it very accessible to most clinicians.

The discussion is exhaustive, surely, all patients should have had urinalysis (line 215-216) to get more information on this aspect. Finally, at line 251 the Author should specify what reduction in contamination he means: only faecal contamination? If not, since no urinalysis was done, it is complicated to state this concept.

Line 224: the author states that stent migration is not common, but 3/17 is about 17-18% of cases. Maybe the definition “uncommun” needs to be changed?

Nowhere in the text is the patient's tolerance to the stent described: has an Elizabethan collar or other deterrent ever been used to avoid attempted removal with the mouth? If so, it must be described.

Some problems reported in the introduction remain unsolved:

- if in most cases the subjects remain incontinent during the permanence of the stent and in some cases there is migration, does the statement of reduced hospitalization remain valid?

- If the bacteriological examination of the urine before and after stent application has not been performed, how can it be shown that the stent, compared to indwelling catheterization, reduces the frequency (and / or severity) of infections?

Overall, the paper is well written, but important considerations are missing in relation to the fact that neoplastic masses sometimes result in complete crushing of the soft catheter, so I recommend expanding on this in the discussion as a limitation of the application of such temporary stents.

I believe that the observations listed above require substantial changes to the text in order for the proposed paper to be eligible for publication.

Author Response

 I appreciated the care taken in review of the manuscript and have provided an explanation and changes for the concerns raised in the original submission. 

Reviewer 1

Comments and Suggestions for Authors

The article deals with a particularly important topic in palliative medicine: it is not always possible to perform surgery on the urethra or bladder at the urethral orifice and these are life treating conditions.

The title is appropriate for the paper and the abstract summarizes well the topic that will be addressed.

As a suggestion I make to the author, the keywords should be words that do not already appear in the title, so that we can broaden the searchability of the article.

Response: Thank you for the recommendation.  “Dog” and “urethral stent” were removed; “urinary catheter” and “transitional cell carcinoma” were added.

The introduction states that the use of the stent for the control of urethral obstruction, compared to indwelling catheterization, is preferable as it reduces the risk of infection and hospitalization but, the available stents are of limited use due to the high cost. Therefore, the author proposes an inexpensive and easy-to-make stent model.

The results are clearly understood. At line 183 it is stated that the analysis was done on “free catch” urine, so this method makes the result non completely reliable.

Response: I agree that cystocentesis is the standard collection method for urine culture.  However, based on results of a recent study (Evaluation of different sampling methods and criteria for diagnosing canine urinary tract infection by quantitative bacterial culture, The Veterinary Journal 2016;216:168–173), voided samples with high colony counts are also appear reliable.  For dogs with clinical signs, the positive predictive value and negative predictive values for voided urine for culture were 96% and 95% respectively. To assist the reader in interpreting the reliability of the culture results, the CFUs for the infections were provided and the reference of the above paper was sited.

The materials and methods section clearly describes the technique and principles behind the choice of a temporary stent, and certainly the availability of the material used makes it very accessible to most clinicians.

The discussion is exhaustive, surely, all patients should have had urinalysis (line 215-216) to get more information on this aspect.

Response: Because cases were managed by a variety of clinicians and 12 of 17 dogs had terminal disease, post-stent urinalyses were rarely performed.   In most cases, I assume that the primary clinicians determined that it would provide little additional information beyond that gathered from the clinical signs. Although the urinalysis is a sensitive test for kidney and metabolic disease, the urinalysis is a poor discriminator for lower urinary tract pathology and is affected by diet, time of feeding, degree of hydration, method of collection, medication, and other concomitant urinary diseases (Evaluation of modified Wright-staining of urine sediment as a method for accurate detection of bacteriuria in dogs. J Am Vet Med Assoc. 2004 Apr 15;224(8):1282-9; Evaluation of urinalyses from untreated adult cats with lower urinary tract disease and healthy control cats: predictive abilities and clinical relevance. J Feline Med Surg. 2013 Dec;15(12):1086-97).  When infection was considered in this study of temporary stents, the urine was cultured.

Finally, at line 251 the Author should specify what reduction in contamination he means: only faecal contamination? If not, since no urinalysis was done, it is complicated to state this concept.

Response: The sentence is qualitied in the beginning with the adjective “Potential advantage.”  It was not intended to be a reflection of the study dogs, but helpful to clinicians based on studies in the literature (references 1 to 4). In line 251, fecal and environmental contamination was intended. A dog’s household is likely to contain fewer antibiotic-resistant urinary pathogens per area than a veterinary ICU setting.  Because it is unknown if the dog’s house has fewer bacteria, “fecal” was added to the manuscript for clarification even though more than fecal contamination was considered. In addition, “closed collection system” was added to the description of indwelling catheters for clarification.  These changes should improve the readers’ understanding and evaluation of the study.

Line 224: the author states that stent migration is not common, but 3/17 is about 17-18% of cases. Maybe the definition “uncommun” needs to be changed?

Response; the word uncommon was removed. Thank you.

Nowhere in the text is the patient's tolerance to the stent described: has an Elizabethan collar or other deterrent ever been used to avoid attempted removal with the mouth? If so, it must be described.

Response: the stent was well tolerated; no dogs required an E-collar or other deterrent apparatus.  Thank you for the clarification.  This information was added to the results section of the manuscript.

Some problems reported in the introduction remain unsolved:

- if in most cases the subjects remain incontinent during the permanence of the stent and in some cases there is migration, does the statement of reduced hospitalization remain valid?

Response: In the introduction, hospitalization refers to indwelling catheters which are usually connected to a closed-collection system.  In our hospital and in the published literature (Evaluation of open versus closed collection systems…J Am Vet Med Assoc 2010;237:187–190), dogs are often hospitalized until the catheter is removed.  Temporary stent placement is an outpatient procedure with patients discharged the same day. Even with recheck evaluations, hospitalization would be shorter. Therefore, the statement should be appropriate.  The introduction was modified to clarify that temporary stent placement is an outpatient procedure.

- If the bacteriological examination of the urine before and after stent application has not been performed, how can it be shown that the stent, compared to indwelling catheterization, reduces the frequency (and / or severity) of infections?

Response: I could not find a place in the manuscript that states that the use of temporary stents compared to indwelling catheters reduces urinary tract infections. The paper states that, “Further studies are needed to determine the risk of urinary tract infection in dogs with temporary stents.”  In another paragraph that references indwelling catheters, it states that, Potential advantages…include…reduced…contamination.”  It is not stated as a fact or that the manuscript provides evidence for it.  In the discussion it states, “It is unknown if urinary infection rates would be lower for a temporary stent in a dog residing in its home where bacteria populations are potentially fewer and less virulent compared to maintaining an indwelling urinary catheter in a veterinary intensive care setting.” In the paragraph on limitations, it is written that, “...limited the study’s ability to assess infection rate…”

Overall, the paper is well written, but important considerations are missing in relation to the fact that neoplastic masses sometimes result in complete crushing of the soft catheter, so I recommend expanding on this in the discussion as a limitation of the application of such temporary stents.

Response: Compression and ingrowth of tumor is reported with the expanding metallic stents.  Using the search words urinary catheter, urethral tumor (cancer), prostatic tumor (cancer), transitional cell carcinoma, and indwelling catheter; no papers identified that catheters were compressed by the tumor.  No cases in this study identified catheter compression.  If the reviewer were aware of a report that was overlooked, providing a reference would be appreciated.  It would be great addition to the discussion. Thank you for bringing it up.  Maybe catheter compression was not seen in this study because the catheter was for temporary use and not long term use.

I believe that the observations listed above require substantial changes to the text in order for the proposed paper to be eligible for publication.

Reviewer 2 Report

This manuscript describes the use of a low-cost temporary Urethral Stents in The Management 2 of Malignant and Non-Malignant Urethral Diseases in Dogs. It is a interesting manuscript, and this reviewer has only few comments. Please see below:

  1. Any inclusion/exclusion criteria were applied for the evaluation of medical records. Please, describe.
  2. Not sure if I lack this information; however, did the authors described the reference for this method? Because if is a new technique, ethic statement and informed consent it will be necessary. If is a well-recognized technique, accept to treat patients, please, provide references.

Author Response

I appreciated the care taken in review of the manuscript and have provided an explanation and changes for the concerns raised in the original submission. 

Reviewer 2.

Comments and Suggestions for Authors

This manuscript describes the use of a low-cost temporary Urethral Stents in The Management of Malignant and Non-Malignant Urethral Diseases in Dogs. It is a interesting manuscript, and this reviewer has only few comments. Please see below:

  1. Any inclusion/exclusion criteria were applied for the evaluation of medical records. Please, describe.

Response: There was no additional inclusion or exclusion criteria applied to the evaluation of the records in relation to the stent.  The records were not evaluated for treatment of the primary urethral disease. This information was added to the manuscript for clarification.

  1. Not sure if I lack this information; however, did the authors described the reference for this method? Because if is a new technique, ethic statement and informed consent it will be necessary. If is a well-recognized technique, accept to treat patients, please, provide references. Response: Placement of indwelling catheters and stents is not new (although there are no prospective studies proving their benefit).  However, modification of the catheter/stent and its novel attachment are new and was developed by the author. There are no previous reports of this procedure.  In our hospital, all treatments require informed consent.  The owner consent form for patient care was sent to the journal at the time of submission.  Because the device was not being tested in a prospective study, but deemed important for preserve patient health, a separate or stent-specific consent form was not needed in the care of patients presenting with urgent disease.

Reviewer 3 Report

Reviewer comments for manuscript ID vet sci-1541948 ‘Evaluation of Temporary Urethral Stents in The Management of Malignant and Non-Malignant Urethral Diseases in Dogs’

General Comments

It is an innovative work on the placement of temporary and inexpensive stents for managing urethral stenosis and urinary obstruction in dogs. It is an important work that can be adopted in veterinary practices all over the world as this is a very common clinical problem in dogs. The manuscript is well written with detailed follow up of the cases despite being a retrospective study. These studies are very important as they comprehensively inform about the outcomes of a technique and procedure and highlight the limitations. I have very minor corrections and suggestions to the author to undergo following which I recommend the publication of the manuscript.

Specific Comments

Line 129: Please check this sentence ’This dog displayed normal urinary continence’. I think should be normal urination. Please clarify.

Lines 210-11: Please reword this sentence ‘, it is not question of if the urinary tract will become colo- nized with bacteria but when’ as ‘ it is critical to know the time when the urinary tract will be colonised by bacteria rather than the colonization itself’

Author Response

I appreciated the care taken in review of the manuscript and have provided an explanation and changes for the concerns raised in the original submission. 

General Comments

It is an innovative work on the placement of temporary and inexpensive stents for managing urethral stenosis and urinary obstruction in dogs. It is an important work that can be adopted in veterinary practices all over the world as this is a very common clinical problem in dogs. The manuscript is well written with detailed follow up of the cases despite being a retrospective study. These studies are very important as they comprehensively inform about the outcomes of a technique and procedure and highlight the limitations. I have very minor corrections and suggestions to the author to undergo following which I recommend the publication of the manuscript.

Specific Comments

Line 129: Please check this sentence ’This dog displayed normal urinary continence’. I think should be normal urination. Please clarify.

Response: the wording was changed to “normal urination.”  Thank you.

Lines 210-11: Please reword this sentence ‘, it is not question of if the urinary tract will become colo- nized with bacteria but when’ as ‘ it is critical to know the time when the urinary tract will be colonised by bacteria rather than the colonization itself’

Response: Thank you for the suggestion.  I made changes to the manuscript to reflect this suggestion.

Round 2

Reviewer 1 Report

The major remarks that I have made have had an adequate response. I thank the author for the promptness in his answers. The paper can be accepted, in my opinion, in its present form